# Soil Consolidation Analysis in the Context of Intermediate Foundation as a New Material Perspective in the Calibration of Numerical–Material Models

Grzegorz Kacprzak and Mateusz Frydrych *

Civil Engineering Faculty, Warsaw University of Technology, 00-637 Warszawa, Poland;
grzegorz.kacprzak@pw.edu.pl
* Correspondence: mateusz.frydrych.dokt@pw.edu.pl; Tel.: +48-22-2346515

**Abstract:** This paper presents the authors' research results from an analysis of intermediate foundations as well as slab and pile foundations in the context of soil consolidation. Looking at soil as a building material that changes its properties over time is very important from the point of view of the safety of construction, implementation, and operation of building structures. In addition, soil can be parameterized in such a way as to accurately describe its possible behavior under service loading. Of great interest is the phenomenon of consolidation, which is based on the reduction of soil volume over time under constant loading. This study explores existing piles and replicates soil conditions to understand individual and grouped pile behavior in combined pile–raft foundations (CPRF). To assess pile settlement from primary and secondary consolidation phases, 13 field measurements on concrete columns in gyttja clay were conducted. Analyzing data from these tests allowed engineers to accurately calibrate a numerical model. This calibrated model was instrumental in designing high-rise buildings, ensuring stability and safety. This study emphasizes the importance of understanding soil behavior, particularly consolidation phenomena, in optimizing foundation design and construction practices.

**Keywords:** consolidation; CPRF; numerical analysis; gyttja; deformable soil





## 1. Introduction

Prior to exploring soil consolidation from a different angle with slab and pile foundations, it is vital to view soil as a material possessing remarkable qualities, such as alterations in its structure and deformable properties over time and under continuous loading. In practice, this manifests itself as settlement, which occurs during the operation of the project. What are combined pile–raft foundations and how does the issue of consolidation change for systems related to the number of piles and their combination? In general, building structures are usually founded on two basic types of foundations:

- Foundations founded directly (footings or slabs) on load-bearing and low-deformation soils;
- Indirect (deep) foundations based on lower layers of load-bearing soils, in the form of a certain number of piles or piles connected to the structure by a cap (grate, footing, slab) to transfer loads.

In Poland, direct foundations are designed based on the guidelines of PN-81/B-03020 [1], the updated version of which incorporates the recommendations of the European standard PN-EN 1997-1:2008 EUROKOD 7—PART 1 [2] and PN-EN 1997-2:2009 Eurocode 7—Part 2 [3], and AMENDED PN-81/B-03020 [4]. On the other hand, foundations on pile foundations are designed according to the recommendations of PN-83/B-02482 [5] or the mentioned two parts of Eurocode 7 [2,3]. When the strength or deformation parameters of the soil are insufficient for the direct foundation of a structure, it is worth seeking a rational and optimal way of

transferring loads to the ground. The use of an indirect foundation on piles in such a situation means that the resistance of the soil mobilized under the mesh will be ignored [6]. Therefore, when ground conditions allow it, i.e., the subsoil is sufficiently undeformable and at least to a limited extent load-bearing, it is possible to analyze the inclusion of the soil under the pile, i.e., to combine a direct foundation with a deep foundation in order to mobilize both types of foundations simultaneously [7,8]. Such a combination of two foundation methods, popularly referred to in the literature as a combined slab–pile foundation (German: KPP—Kombinierte Pfahl-Plattengründungen; CPRF—combined pile–raft foundation) or simply mixed foundation (French: Fondations Mixtes), allows minimizing the number and length of deep foundation elements, such as piles, resulting from the traditional solution [9]. In the remainder of this paper, the term slab–pile foundation (CPRF) will be used.

The period of the last twenty-five years can be described as a time of significant development of slab–pile foundations. During this period, many new methods of analyzing the behavior of CPRFs were developed, using increasingly widely available tools for modeling the cooperation of the ground medium with the structure, data from observations of settlements of realized structures, as well as all kinds of experiments on a natural and laboratory scale.

The first analyses of the cooperation of slab–pile foundations with the soil medium were initiated as early as the late 1970s and early 1980s. In France, the information obtained at the stage of dimensioning this type of foundation was compiled in a study by the LCPC (Laboratoire Central des Ponts et Chaussees) [10] presenting a design method verified and confirmed in practice. The study, including the then-current design methods for slab-and-pile foundations, appeared in the form of the 2001 LCPC Report [11] edited by Serge Borel, entitled "Comportement et dimensionnement des fondations mixtes" (Behavior and dimensioning of mixed foundations).

In Germany, until 2001, a mainly traditional design approach was practiced in terms of the foundation of buildings. Foundations were designed as direct or indirect in accordance with the recommendations of DIN 1054 [12] (along with other referenced standards). The gap between the two types of foundation was bridged with the DIBt (Deutsches Institut für Bautechnik) Instruction issued in 2002 under the title "Richtlinie fur den Entwurf, die Bemessung und den Bau von Kombinierte Pfahl-Plattengründungen" [13] (Guide to the Design and Construction of Combined Slab–Pile Foundations) providing guidelines for the analysis and application of the new type of foundation.

The most up-to-date set of slab–pile foundation design guidelines recommended for use by designers worldwide is the guide entitled "ISSMGE combined pile-raft foundation guidelines", first published in May 2012 by the Deep Foundation Technical Committee No. 212 in the form of an Annual Technical Report [14]. The final version of the June 2013 guide [15] prepared by Rolf Katzenbach and Deepankar Choudhury is a compendium of knowledge on the design, execution, and monitoring of slab–pile foundations.

In this article, the authors take a closer look at aspects of soil consolidation in light of the design of slab–pile foundations, which is crucial for the correct calibration of numerical models for analyzing deep foundations. This kind of perspective allows for optimal foundation design by understanding how to properly describe the soil material mathematically.

## 2. Materials and Methods—Background Information

As written in the introduction, the term slab–pile foundation refers to the combination of a direct foundation and a deep foundation. As a result of this connection, both the soil lying directly under the direct foundation and the soil around the deep foundation are included in the load transfer. Figure 1 shows a schematic comparison of the working mechanisms of different types of foundations and their interaction with the soil medium.

According to [9], a slab-and-pile foundation should be regarded as a complex geotechnical structure, in which the global load of the foundation is transferred to S the foundation's components, i.e., the soil under the slab and the piles, with mutual interactions (Figure 2).

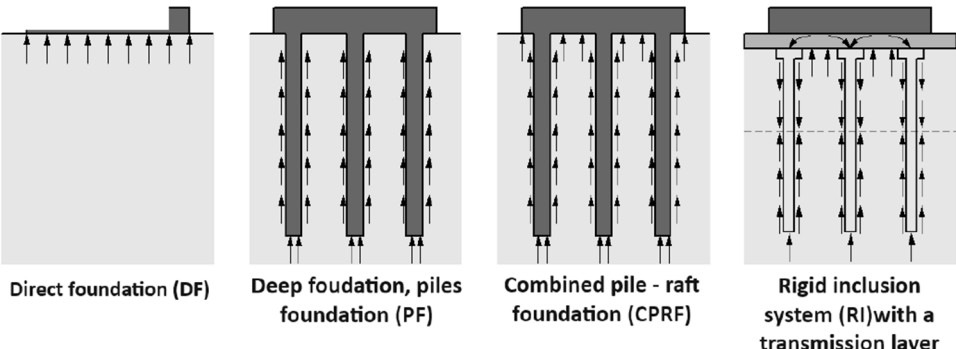

**Figure 1.** The most common types of foundations. Adapted from [16].

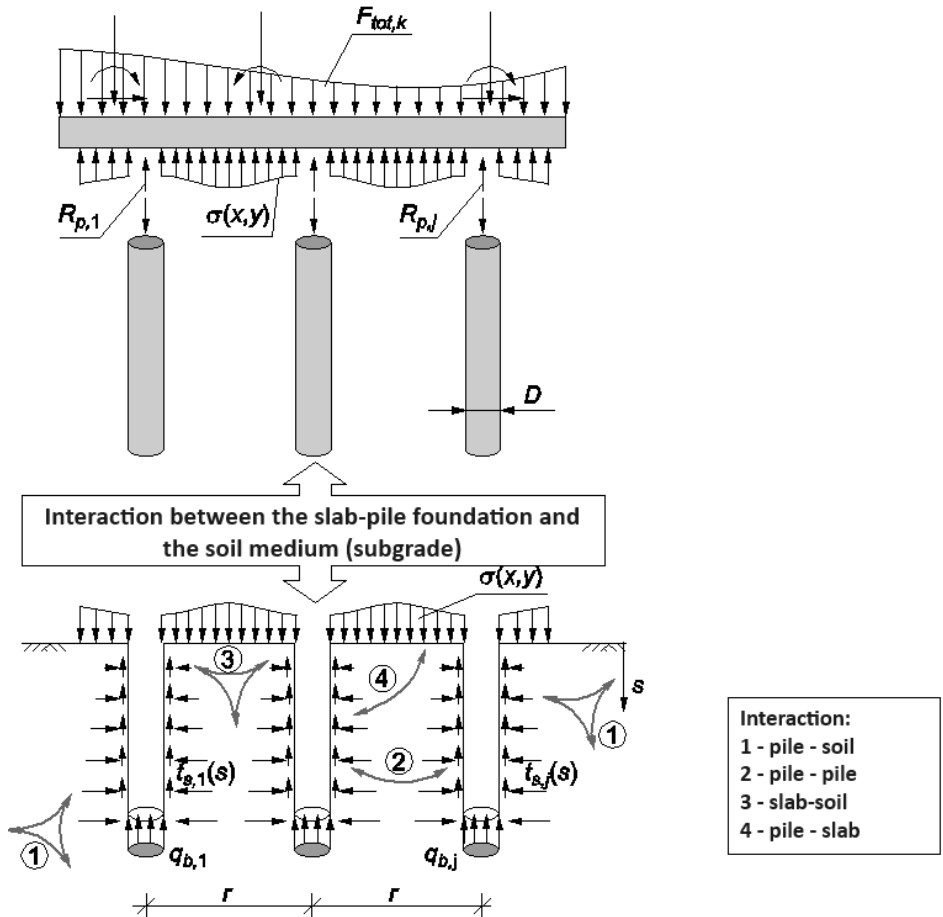

**Figure 2.** Slab–pile foundation (CPRF) as a complex geotechnical structure. Interaction of the elements of a slab–pile foundation according to Hanisch et al. [13].

Hanish and others [13] make the operating range of a slab–pile foundation dependent on the soil conditions under a direct foundation and on the value of the load that the piles take on in the form of a factor $\alpha$CPRF, defined as the quotient of the load carried by the piles $\sum_{j=1}^{n} R_{p,j}(s)$, and the total load on the foundation $F_{\text{tot},k}(s)$, where s denotes foundation settlement:

$$\alpha_{\text{CPRF}}(s) = \frac{\sum_{j=1}^{n} R_{p,j}(s)}{F_{\text{tot},k}(s)} \tag{1}$$

The $\alpha$CPRF coefficient can take values in the range of $0 \div 1$, where a value of 0 indicates a direct foundation and a value of 1 indicates a pile foundation without slab participation (no slab-to-ground contact). Figure 3 shows the course of the dependence of

the αCPRF coefficient on the ratio of the settlement of the slab–pile foundation sCPRF to the settlement of the direct foundation sFI (assuming identical surfaces of the two foundations and identical loading). The graph allows assessing the suitability of the slab–pile foundation to reduce settlement. Hanish and others [13] assume that a slab–pile foundation can be said to be a slab–pile foundation when αCPRF ≤ 0.9 and when there is soil under the direct foundation whose deformation characteristics are at least equal to 10% of the deformation characteristics of the soil in which the pile bases are embedded.

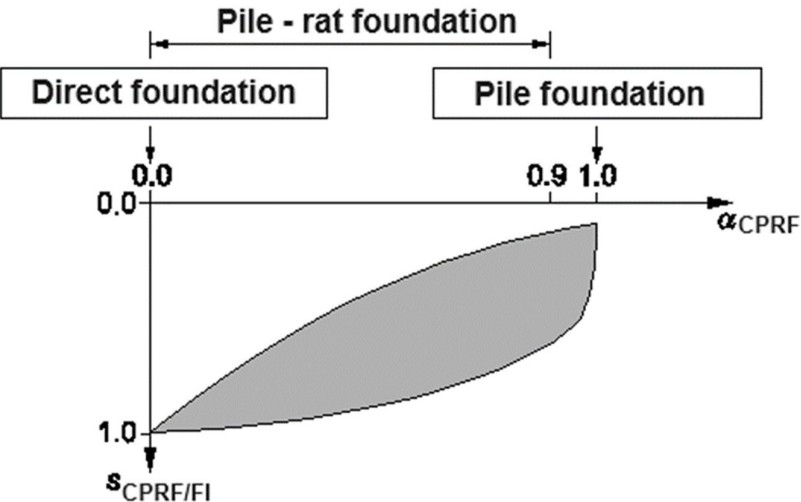

**Figure 3.** The course of the dependence of the coefficient αCPRF as a function of the ratio of the settlement of the CPRF foundation consisting of a direct foundation (e.g., slab) and a group of piles sCPRF to the settlement of the direct foundation sFI (sCPRF/sFI means the degree of reduction in the settlement of the direct foundation after the introduction of piles into the foundation).

As it is possible to see, CPRF is used for deformable soils where high demands are made to limit deformation over time. A good example is bridge structures, whose requirements for vertical displacement are very demanding. This is because it is unacceptable to achieve vertical displacement, as a traffic accident could occur during operation. Borel [11] in his work gives examples of several engineering structures in France and around the world (Italy, Finland, Mexico), including bridges, where slab–pile foundations were used. The first such structure in France was a bridge over the RN 138 national road south of Rouen, built in 1981.

In the case of buildings where the foundation cannot be realized with footings or benches, the load on the ground is transferred through a foundation slab, the width of which usually exceeds 15–20 m. In such a situation, the ultimate limit state condition is usually met, but it is possible to exceed the permissible settlements. Hence, in order to reduce vertical displacements, it is necessary to additionally use deep foundations to a limited extent. For civil engineering structures, for which the width of the direct foundation varies between 2 and 10 m, the use of a deep foundation is usually associated with the need to meet the ultimate limit state condition [17]. Experience shows that when sizing foundations of small width (e.g., footings), the limit resistance of the subsoil beneath the foundation is most important, while when sizing foundation slabs of large size, the value of allowable settlement is decisive.

In the context of the presented opportunities for slab–pile foundations, the authors will try to answer the question of the consolidation process in the calibration of numerical models and the operation of civil structures.

### 3. Effect of Soil Consolidation on the Total Settlement of a Single Pile

As in the case of direct foundation on a foundation slab, in the case of foundation on piles, the total settlement should take into account all stages of settlement, i.e., immediate settlement and settlement from primary (filtration) and secondary (creep) consolidation.

In order to verify the total value of pile settlement caused by staggered primary and secondary consolidation, bearing in mind that the different phases of soil settlement interpenetrate each other, the results of 13 field measurements of static loads on concrete columns made in the gyttjas of the *Rynna Żoliborska* in Poland, sunken with the base in load-bearing sandy loams/clayey sands, were analyzed. Complementary to 1:1 scale monitoring of building structures, model studies can be used. The model studies presented in this section were performed to analyze the settlement over time of slab-on-grade foundations. The authors focused on presenting an experimentally verified empirical formula for predicting the course of consolidation of the foundation of real-scale buildings. An additional application goal was to indicate the quantitative effect of piles on the increase in bearing capacity and reduction in settlement of slab–pile foundations. The authors prepared laboratory-scale test sites with the interrelationships of the building elements of the CPRF, which were tested in situ and in ground conditions corresponding to the natural ones from the study area. The detailed description of the model tests is quite extensive and is within the scope of another study. From the perspective of this article, it is necessary to use the results themselves. The model tests performed were prepared to show total pile settlements over time. The estimation of the final settlement of the column $s_c = s\infty$, as in the case of estimating the settlement of a direct foundation, consisted of selecting the parameters of the Meyer function ($D_e$, $p$, $\alpha$, and $s\infty$) to match the results of the settlements measured during the test loading of the column in the field.

According to the company's own oedometric studies [18] carried out on reconstituted sand and silt samples of different consistencies, the proportion of settlement due to secondary consolidation $s_e$, as part of the settlement that may occur during the operational stage of the structure, to the total settlement $s_c$, can be considered constant and independent of the value of stress in the soil medium of the same consistency. Under this assumption, the ratio of the settlement of the column at the operational stage $s_e$ to the total settlement of the column $s_c$ was determined by the increments of settlement caused by the change in the load on the column head from 500 (kN) to 600 (kN) ($\Delta\sigma = $ const $ = 100$ (kN)). The settlement of the column at the $s_e$ operation stage was calculated as the difference between the total settlement $s_c$ and the settlement observed after the contractual settlement stabilization time, i.e., after about 30 min for most columns. All data was presented in Tables 1–3.

**Table 1.** Meyer function parameters for columns 1, 3, 4, 10, 12, 13, 14, and 17.

| Column | 1 | 3 | 4 | 10 | 12 | 13 | 14 | 17 |
|---|---|---|---|---|---|---|---|---|
| $s_\infty$ | 2.73 | 1.23 | 1.99 | 1.05 | 1.90 | 1.84 | 1.10 | 1.00 |
| $D_e$ | 3.01 | 5.00 | 1.39 | $1.00 \times ^{-11}$ | 4.99E | 3.80 | 3.65 | 2.85 |
| $p$ | $7.39 \times 10^{-1}$ | $4.19 \times 10^{-1}$ | $1.00 \times 10^{-3}$ | $1.05 \times 10^1$ | $6.19 \times 10^{-1}$ | $2.34 \times 10^{-1}$ | $3.72 \times 10^{-1}$ | $3.10 \times 10^{-1}$ |
| $\alpha$ | $1.00 \times 10^{-6}$ | 1.34 | 1.41 | 4.82 | 4.03 | $8.19 \times 10^{-1}$ | 2.10 | $1.00 \times 10^{-6}$ |

**Table 2.** Meyer function parameters for columns 7, 8, 9, 11, and 16.

| Column | 7 | 8 | 9 | 11 | 16 |
|---|---|---|---|---|---|
| $s_\infty$ | $5.32 \times 10^{-1}$ | $7.10 \times 10^{-1}$ | 1.01 | $6.17 \times 10^{-1}$ | $7.77 \times 10^{-1}$ |
| $D_e$ | $4.10 \times 10^{-1}$ | 2.68 | $5.56 \times 10^{-1}$ | 5.00 | $6.92 \times 10^{-1}$ |
| $p$ | $1.00 \times 10^{-3}$ | $3.11 \times 10^{-1}$ | $8.72 \times 10^{-2}$ | $4.99 \times 10^{-1}$ | $6.61 \times 10^{-2}$ |
| $\alpha$ | 1.82 | $1.00 \times 10^{-6}$ | $2.90 \times 10^{-1}$ | 5.00 | $4.80 \times 10^{-5}$ |

**Table 3.** Contribution of column settlement during the operation stage to the total settlement according to Meyer's formula.

| Column | 1 | 3 | 4 | 7 | 8 | 9 | 10 | 11 | 12 | 13 | 14 | 16 | 17 |
|---|---|---|---|---|---|---|---|---|---|---|---|---|---|
| $s_e$ | 0.08 | 0.03 | 0.24 | 0.13 | 0.10 | 0.54 | 0.04 | 0.01 | 0.00 | 0.06 | 0.04 | 0.42 | 0.12 |
| $s_c$ | 2.73 | 1.23 | 1.99 | 0.53 | 0.71 | 1.01 | 1.05 | 0.62 | 1.90 | 1.84 | 1.10 | 0.78 | 1.00 |
| $s_e/s_c$ | 0.03 | 0.02 | 0.12 | 0.25 | 0.14 | 0.53 | 0.04 | 0.01 | 0.00 | 0.03 | 0.03 | 0.54 | 0.12 |

Comparing the settlement of column heads at the end of the load test with the settlement estimated using the consolidation model (Figure 4), there are basically two groups of columns. The first is a group of columns (nos. 13, 12, 14, 10, 1, 3, and 11) for which the share of exploitation settlement is small or close to 0, and the second group of columns for which the settlement at the exploitation stage is at the level of 12–25% (17, 4, 7, and 8).

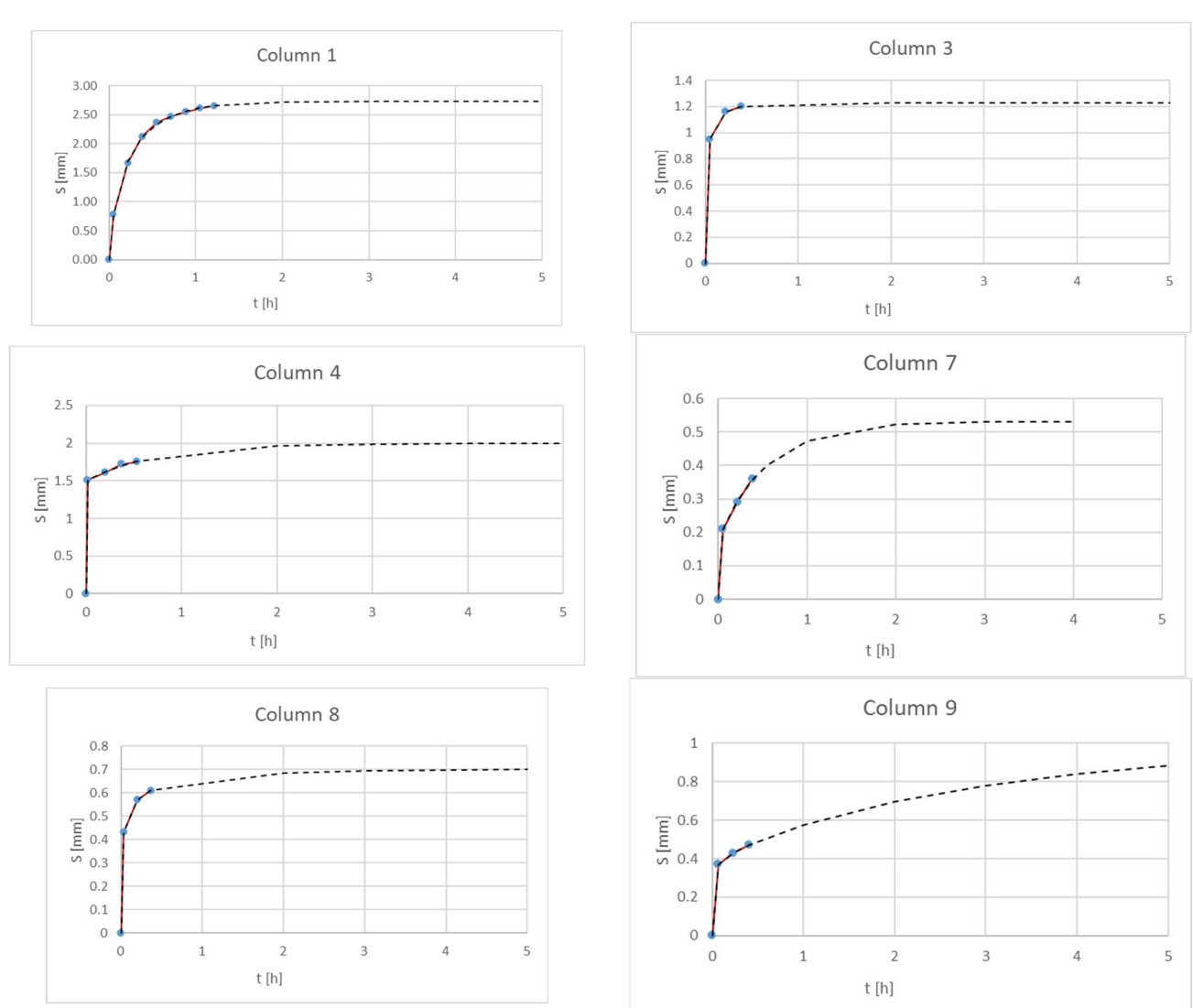

**Figure 4.** *Cont.*

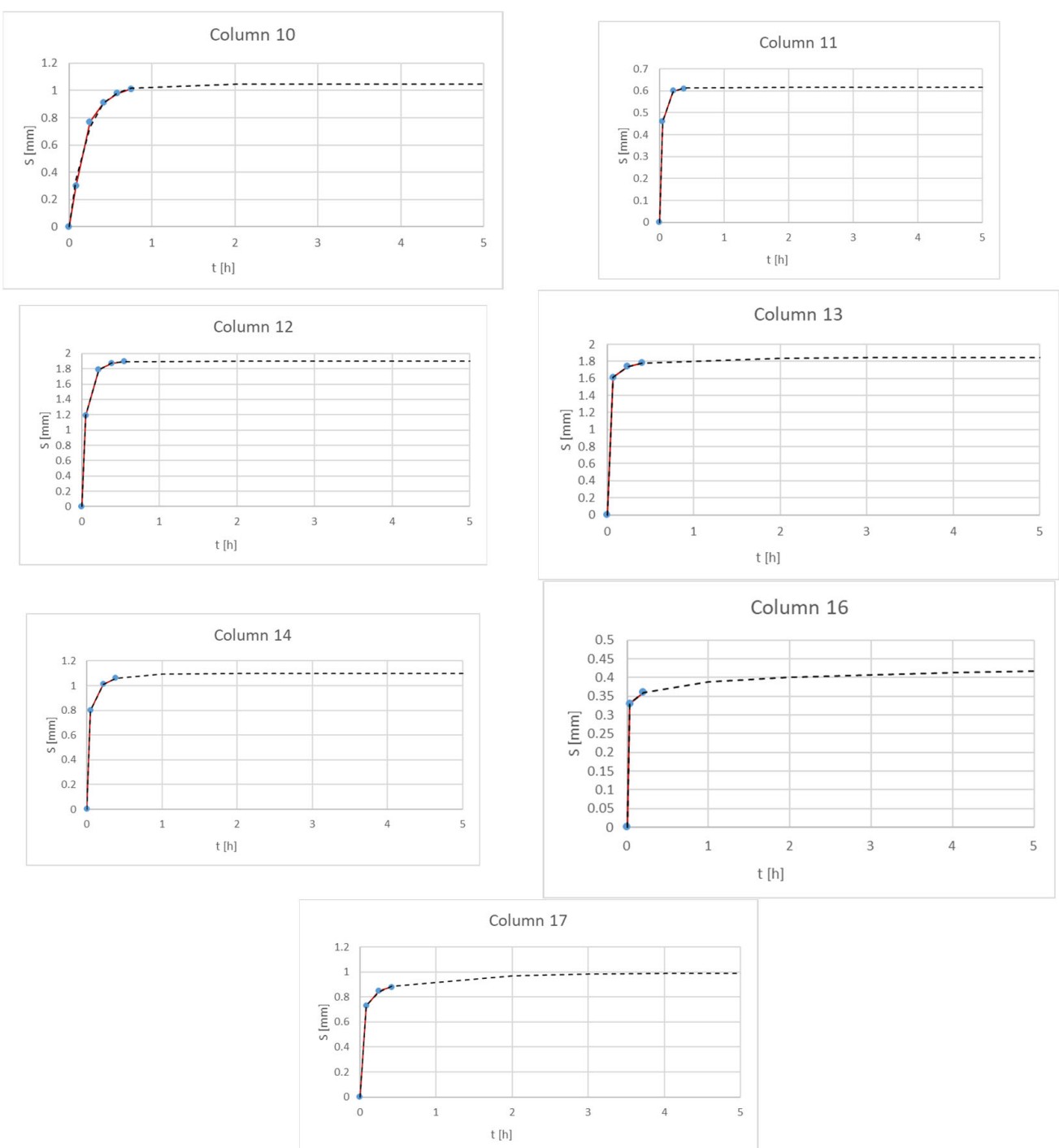

**Figure 4.** Fitting the Meyer curve (dashed line) to the measurement results (blue points).

## 4. Consolidation Settlements

In order to estimate the settlement stabilization time and determine the final settlement, the consolidation curves from our own model tests were analyzed by fitting a Meyer curve to them (see Figures 5–8). Model studies described in a separate publication by the authors were presented to analyze settlement over time of slab-on-grade foundations [19].

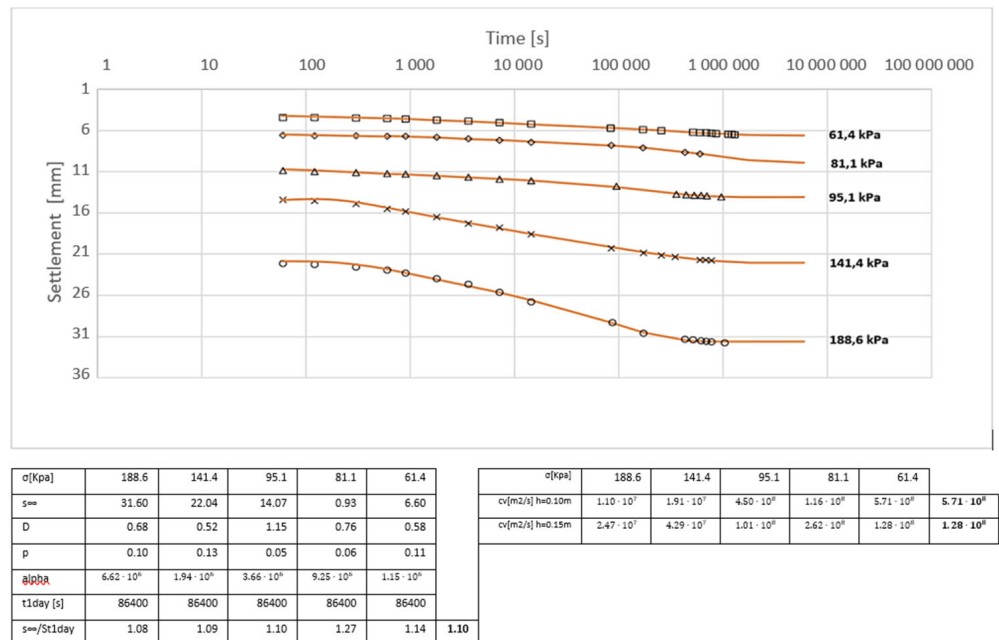

| σ[Kpa] | 188.6 | 141.4 | 95.1 | 81.1 | 61.4 | |
|---|---|---|---|---|---|---|
| s∞ | 31.60 | 22.04 | 14.07 | 0.93 | 6.60 | |
| D | 0.68 | 0.52 | 1.15 | 0.76 | 0.58 | |
| p | 0.10 | 0.13 | 0.05 | 0.06 | 0.11 | |
| alpha | $6.62 \cdot 10^6$ | $1.94 \cdot 10^6$ | $3.66 \cdot 10^6$ | $9.25 \cdot 10^6$ | $1.15 \cdot 10^6$ | |
| t1day [s] | 86400 | 86400 | 86400 | 86400 | 86400 | |
| s∞/St1day | 1.08 | 1.09 | 1.10 | 1.27 | 1.14 | **1.10** |

| σ[Kpa] | 188.6 | 141.4 | 95.1 | 81.1 | 61.4 | |
|---|---|---|---|---|---|---|
| cv[m2/s] h=0.10m | $1.10 \cdot 10^7$ | $1.91 \cdot 10^7$ | $4.50 \cdot 10^8$ | $1.16 \cdot 10^8$ | $5.71 \cdot 10^8$ | $\mathbf{5.71 \cdot 10^8}$ |
| cv[m2/s] h=0.15m | $2.47 \cdot 10^7$ | $4.29 \cdot 10^7$ | $1.01 \cdot 10^8$ | $2.62 \cdot 10^8$ | $1.28 \cdot 10^8$ | $\mathbf{1.28 \cdot 10^8}$ |

**Figure 5.** Plate without piles. Approximation of selected compressibility curves with Meyer curve to determine total consolidation settlement. Contribution of total settlement to settlement after 24 h. Values of $c_v$ for $H_{dr}$ = 10 and 15 cm.

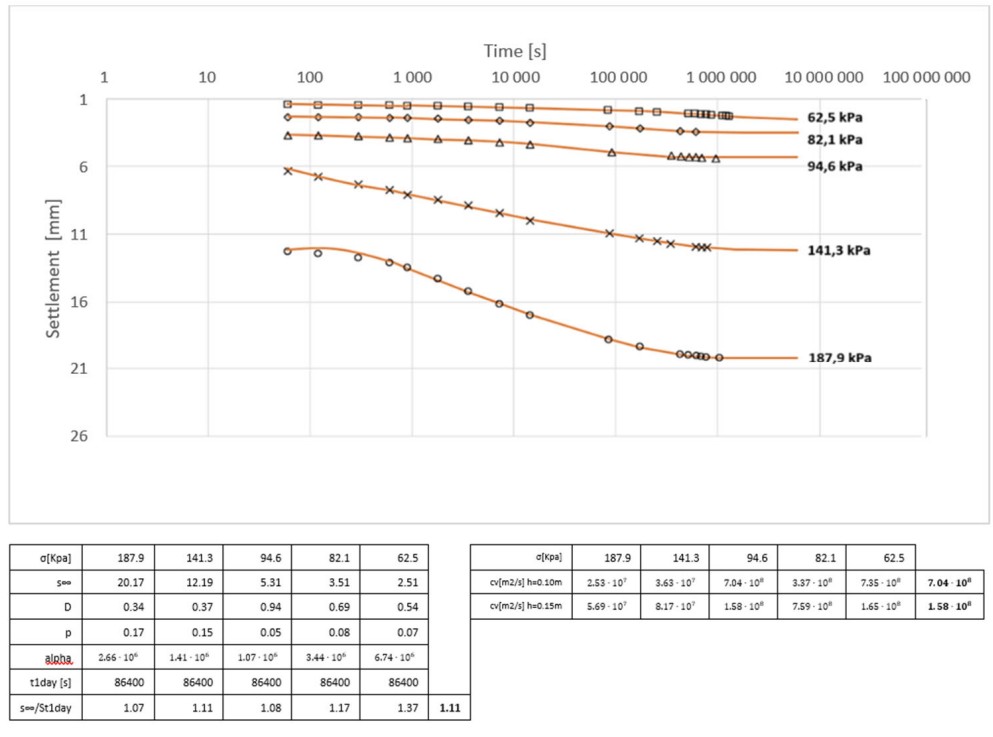

| σ[Kpa] | 187.9 | 141.3 | 94.6 | 82.1 | 62.5 | |
|---|---|---|---|---|---|---|
| s∞ | 20.17 | 12.19 | 5.31 | 3.51 | 2.51 | |
| D | 0.34 | 0.37 | 0.94 | 0.69 | 0.54 | |
| p | 0.17 | 0.15 | 0.05 | 0.08 | 0.07 | |
| alpha | $2.66 \cdot 10^6$ | $1.41 \cdot 10^6$ | $1.07 \cdot 10^6$ | $3.44 \cdot 10^6$ | $6.74 \cdot 10^6$ | |
| t1day [s] | 86400 | 86400 | 86400 | 86400 | 86400 | |
| s∞/St1day | 1.07 | 1.11 | 1.08 | 1.17 | 1.37 | **1.11** |

| σ[Kpa] | 187.9 | 141.3 | 94.6 | 82.1 | 62.5 | |
|---|---|---|---|---|---|---|
| cv[m2/s] h=0.10m | $2.53 \cdot 10^7$ | $3.63 \cdot 10^7$ | $7.04 \cdot 10^8$ | $3.37 \cdot 10^8$ | $7.35 \cdot 10^8$ | $\mathbf{7.04 \cdot 10^8}$ |
| cv[m2/s] h=0.15m | $5.69 \cdot 10^7$ | $8.17 \cdot 10^7$ | $1.58 \cdot 10^8$ | $7.59 \cdot 10^8$ | $1.65 \cdot 10^8$ | $\mathbf{1.58 \cdot 10^8}$ |

**Figure 6.** Slab with 4 piles. Approximation of selected compressibility curves with Meyer curve to determine total consolidation settlement. Contribution of total settlement to settlement after 24 h. Values of $c_v$ for $H_{dr}$ = 10 and 15 cm.

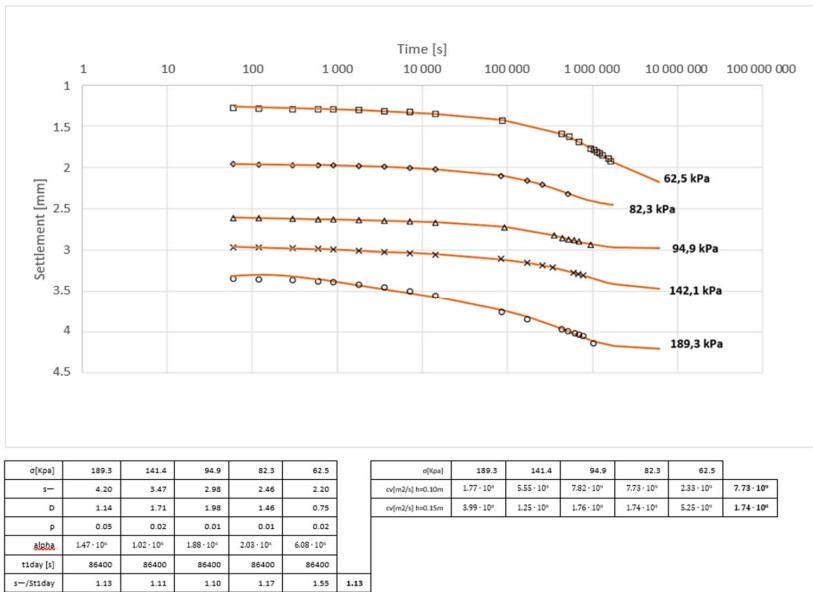

| σ[Kpa] | 189.3 | 141.4 | 94.9 | 82.3 | 62.5 | | σ[Kpa] | 189.3 | 141.4 | 94.9 | 82.3 | 62.5 | |
|---|---|---|---|---|---|---|---|---|---|---|---|---|---|
| s∞ | 4.20 | 3.47 | 2.98 | 2.46 | 2.20 | | cv[m2/s] h=0.10m | $1.77 \cdot 10^8$ | $5.55 \cdot 10^6$ | $7.82 \cdot 10^6$ | $7.73 \cdot 10^6$ | $2.33 \cdot 10^6$ | $7.73 \cdot 10^6$ |
| D | 1.14 | 1.71 | 1.98 | 1.46 | 0.75 | | cv[m2/s] h=0.15m | $3.99 \cdot 10^8$ | $1.25 \cdot 10^8$ | $1.76 \cdot 10^6$ | $1.74 \cdot 10^6$ | $5.25 \cdot 10^6$ | $1.74 \cdot 10^6$ |
| p | 0.05 | 0.02 | 0.01 | 0.01 | 0.02 | | | | | | | | |
| alpha | $1.47 \cdot 10^6$ | $1.02 \cdot 10^6$ | $1.88 \cdot 10^6$ | $2.03 \cdot 10^6$ | $6.08 \cdot 10^6$ | | | | | | | | |
| t1day [s] | 86400 | 86400 | 86400 | 86400 | 86400 | | | | | | | | |
| s∞/St1day | 1.13 | 1.11 | 1.10 | 1.17 | 1.55 | 1.13 | | | | | | | |

**Figure 7.** Slab with 9 piles. Approximation of selected compressibility curves with Meyer curve to determine total consolidation settlement. Contribution of total settlement to settlement after 24 h.

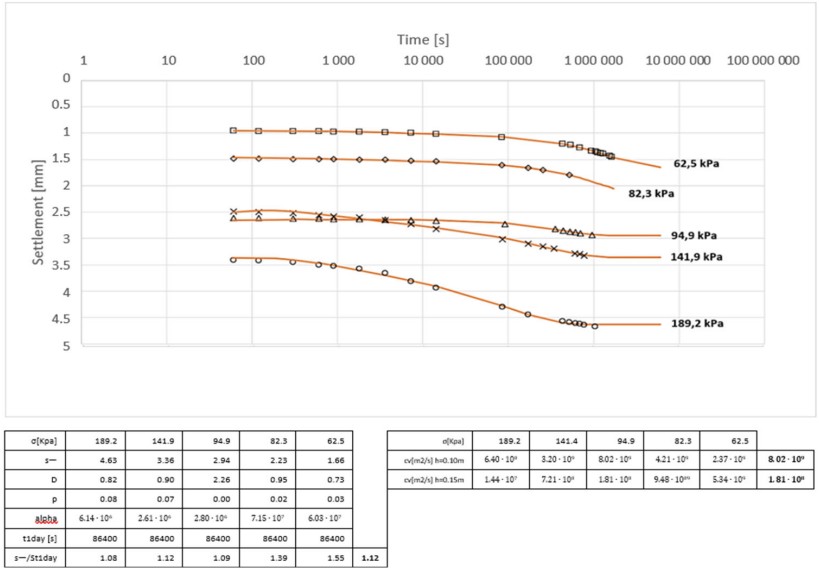

| σ[Kpa] | 189.2 | 141.9 | 94.9 | 82.3 | 62.5 | | σ[Kpa] | 189.2 | 141.4 | 94.9 | 82.3 | 62.5 | |
|---|---|---|---|---|---|---|---|---|---|---|---|---|---|
| s∞ | 4.63 | 3.36 | 2.94 | 2.23 | 1.66 | | cv[m2/s] h=0.10m | $6.40 \cdot 10^8$ | $3.20 \cdot 10^8$ | $8.02 \cdot 10^6$ | $4.21 \cdot 10^6$ | $2.37 \cdot 10^6$ | $8.02 \cdot 10^6$ |
| D | 0.82 | 0.90 | 2.26 | 0.95 | 0.73 | | cv[m2/s] h=0.15m | $1.44 \cdot 10^7$ | $7.21 \cdot 10^8$ | $1.81 \cdot 10^6$ | $9.48 \cdot 10^{10}$ | $5.34 \cdot 10^6$ | $1.81 \cdot 10^6$ |
| p | 0.08 | 0.07 | 0.00 | 0.02 | 0.03 | | | | | | | | |
| alpha | $6.14 \cdot 10^6$ | $2.61 \cdot 10^6$ | $2.80 \cdot 10^6$ | $7.15 \cdot 10^7$ | $6.03 \cdot 10^7$ | | | | | | | | |
| t1day [s] | 86400 | 86400 | 86400 | 86400 | 86400 | | | | | | | | |
| s∞/St1day | 1.08 | 1.12 | 1.09 | 1.39 | 1.55 | 1.12 | | | | | | | |

**Figure 8.** Slab with 16 piles. Approximation of selected compressibility curves with Meyer curve to determine total consolidation settlement. Contribution of total settlement to settlement after 24 h.

The consolidation curves of the test subsoil without and with reinforcement, by making piles under the slab, take the shape of an "inverted S", typical of oedometric tests.

According to the current PN-EN ISO 17892-5:2009, which has its origin in British standards and regulates the methodology and interpretation of oedometric testing, oedometer testing is carried out within 24 h. Similarly, the settlements realized during the in-house model tests after 24 h were considered to correspond to the settlements of the foundation during the construction phase of the building. The settlements mobilized later, i.e., after 24 h (86,400 s) until they reached the value of $s\infty$, were considered to be the settlements mobilized during the building operation phase. This is, of course, a simplification; however, it has a methodological and theoretical basis. The final settlement, approximated by the Meyer curve, was therefore related to the settlement after the conventional time of completion of the construction phase, i.e., after 24 h, thus defining the share of operational settlement (Figures 5–8).

The averaged results from five loading steps (from 62 kPa to 189 kPa as the most common loading range for foundations of typical buildings) for each model (slab and slab with 4, 9, and 16 piles) indicate that the final settlements are 1.10–1.13 times higher than the settlements mobilized in the contractual phase of the structure execution (after 24 h). Relating the values of the subsidence enhancement factors from the contractual phase of construction to estimate the total subsidence, obtained from our own research, to the values recommended in [1,20], and at the same time taking into account the current semi-consolidated state of the tested soil, it should be considered that a large and visible convergence of the results was obtained. The recommendations of ITB [20] and PN-81/B-03020 [1] indicate that for non-cohesive and cohesive soils in the semi-consolidated state, $s_{total} = 1$ to $1.25 \times s_{w\_time\_realisation}$.

The basic differential equation of Terzaghi's consolidation theory shown below, where $c_v$ denotes the consolidation coefficient $c_v = \frac{k \cdot M}{\gamma_w}$, was derived based on the assumption that an equilibrium condition is maintained between the difference in the amount of water flowing in and out of the elementary volume of soil and the change in this volume. The water flow was defined according to Darcy's law, in which the flow velocity is proportionally dependent on the induced hydraulic gradient, through a constant filtration coefficient $k$.

$$\frac{\partial u}{\partial t} = \frac{k E_{oedo}}{\gamma_w} \frac{\partial^2 u}{\partial z^2} = c_v \frac{\partial^2 u}{\partial z^2} \tag{2}$$

Most often, the solution of Terzaghi's equation is presented in the form of the quotient of consolidation settlements $s_t$ mobilized after time $t$, related to total settlements $s_c$ after the consolidation process, defined as the averaged degree of consolidation $U_{av} = s_t/s_c$. Sivaram and Swamee [21] proposed an empirical formula for $U_{av}$ [%]:

$$\frac{U_{av}}{100} = \frac{(4T_v/\pi)^{0.5}}{\left[1 + (4T_v/\pi)^{2.8}\right]^{0.179}} \tag{3}$$

where

$T_v = \frac{c_v t}{H_{dr}^2} t$—time;

$H_{dr}$—filtration path.

In order to estimate the course of consolidation settlement in reality, when designing structures on deformable soils, laboratory tests—oedometric tests—are carried out, allowing the determination of the consolidation coefficient $c_v$ for a given load increment. The Taylor method (square root method) is most often used:

$$c_v = \frac{T_{90} H_{dr90}^2}{t_{90}} \tag{4}$$

where

$T_{90} = 0.848$;

$t_{90}$—time after which $U_{av} = 90\%$ (90% of total consolidation settlement);

$H_{dr90}$—filtration path at $t_{90}$.

And Casagrand (logarithmic method):

$$c_v = \frac{T_{50} H_{dr50}^2}{t_{50}} \tag{5}$$

where

$T_{50} = 0.197$;

$t_{50}$—time after which $U_{av} = 50\%$ (50% of total consolidation settlement);

$H_{dr50}$—filtration path at $t_{50}$.

Originally, in the one-dimensional consolidation equation, $c_v$ is calculated as the product of the filtration coefficient and oedometric modulus, determined for a given load range, related to the volume weight of water $\frac{k E_{oedo}}{\gamma_w}$.

The values of $c_v$ determined by one of the methods given above form the basis for calculating the dimensionless time index $T_v$, needed to estimate the consolidation settlement st after time $t$, at a known (assumed) value of the total settlement mobilized in the consolidation process and with the assumption that the actual filtration path $H_{dr}$ is known. Analogous to oedometric tests, in specially prepared samplers, tests were performed to determine $t_{50}$ according to the Casagrand method for the subsoil without and with reinforcement (using piles). Figures 5–8 show the labeled $c_v$ values determined by the logarithmic method, assuming $H_{dr}$ = 0.10 m as equal to the width of the slab, and $H_{dr}$ = 0.15 m as the total thickness of the soil calculated from the base of the piles to the bottom of the test tank (test tank). The curves in Figures 5–8 showing the variation in settlement as a function of the logarithm of time allowed easy determination of $t_{50}$, and ultimately the determination of the coefficient of consolidation $c_v$.

Figure 9 shows the variation in the consolidation coefficient $c_v$ as a function of load determined assuming $H_{dr}$ = 0.10 m.

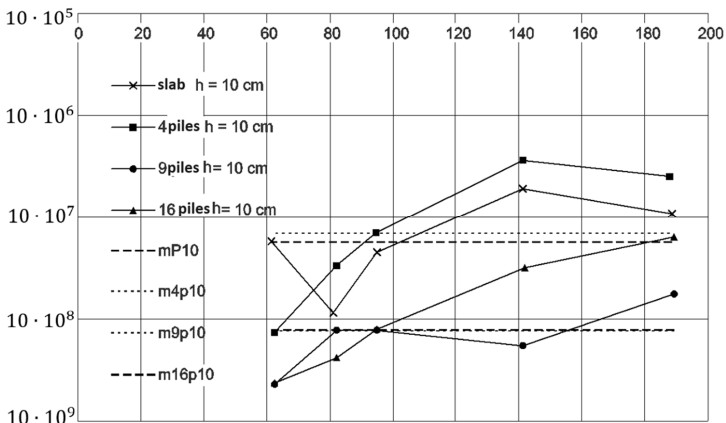

**Figure 9.** Coefficient $c_v$ values for $H_{dr}$ = 10 cm as a function of load. Designations: mP10—median for slab, m4p10—median for 4 piles, m9p10—median for 9 piles, and m16p10—median for 16 piles.

The dependencies of $c_v$ on load for slab and slab–pile foundations (Figure 9), supported by the averaged $c_v$ results of Figures 5–8, indicate two different types of behavior: a stand-alone slab and slab with 4 piles, and a slab supported by 9 and 16 piles. The first group is characterized by $c_v$ consolidation coefficients with average values of $5.71 \times 10^{-8}$ (slab) and $7.04 \times 10^{-8}$ m$^2$/s (slab + four piles). The average $c_v$ values for the second group are an order of magnitude lower: $7.73 \times 10^{-9}$ (slab + 9 piles) and $8.02 \times 10^{-9}$ m$^2$/s (slab + 16 piles). Such observations lead to the conclusion that the consolidation of the subsoil of slab-and-pile foundations, in which the piles are spaced widely (about r/D = 6), proceeds similarly to the consolidation of the subsoil of direct foundations. The CPRF foundation, with piles in close proximity, behaves differently, where consolidation proceeds more slowly.

Analyzing the last two loading steps (142 and 189 kPa) for the slab and the slab with 4 and 16 piles, the $c_v$ takes similar values, close to $1 \times 10^{-7}$ m$^2$/s. This may mean that after exceeding a certain conventional load value, greater than, for example, the value of preconsolidation stress, characteristic of the soil, i.e., when the soil is considered normally consolidated, direct and slab–pile foundations, regardless of the number of piles, settle over time in a similar manner.

As Lambe and Whitman [22] pointed out, determining and selecting the $c_v$ for a specific engineering task is difficult. It should be remembered that the actual settling velocity of a structure's foundation is often two to four times higher than the velocities predicted from the $c_v$ measured using intact samples [23]. At loads of 94, 142, and 189 kPa, the reduction in consolidation settlement using 16 instead of 9 piles is identical or very similar, which means that using more than 9 piles for the studied foundations in the analyzed load range is ineffective.

Using the results of the study of the course of settlement dependence as a function of the logarithm of time, in order to illustrate the variability of the averaged degree of $U_{av}$ consolidation, the consolidation settlement after time t, for each of the five selected loading steps, was related to the total consolidation settlement estimated by the Meyer method. For the first three loading steps, the ground consolidations of slab–pile systems with 9 and 16 piles are very similar. Slab and slab with four piles behave differently. At a load of 95 kPa, we can clearly distinguish between the two types of behavior. The last two loading steps (142 and 189 kPa), discarding the results for the slab with nine piles, indicate that consolidation proceeds in a similar manner regardless of the type of slab–pile foundation. In the case of the nine-pile foundation, analysis of the load–settlement relationship indicated that a lack of smooth entry into the original load path was observed after the reapplication of the load. Interpreting this as a possible measurement error, the results for the slab with nine piles were removed from the above analysis. Then, the selected relations $U_{av(t)}$, in Figure 10, were compared with the consolidation path created from Equation (3). The empirical formula proposed by Sivaram and Swamee [21] indicates a faster completion of the process of dispersion of excess pore water pressure. Hence, for two loading steps (142 and 189 kPa) and three types of CPRF (slab, slab + 4 piles, and slab + 16 piles), the approximation of $U_{av}$ changes as a function of time was carried out using Equation (5). The values of the potentiometric coefficients A, B, and C were determined using nonlinear numerical optimization of in-house results (Table 4). The medians of the potentiometric coefficients for the six selected $U_{av(t)}$ relationships allowed us to write the empirical formula, which is the solution of Terzaghi's one-dimensional consolidation equation, in the form:

$$\frac{U_{av}\%}{100} = \frac{(4T_v/\pi)^A}{\left[1 + (4T_v/\pi)^B\right]^C} \tag{6}$$

with A = 0.437, B = 0.992, and C = 0.443. According to the above conclusions, it is assumed that the consolidation of the subsoil of slab–pile foundations, made of pre-consolidated organic and mineral cohesive soils, for loading greater than 140 kPa, can be determined with some approximation by Formula (5) taking into account the value of the consolidation coefficient $c_v$ determined by oedometric tests, as for direct foundations.

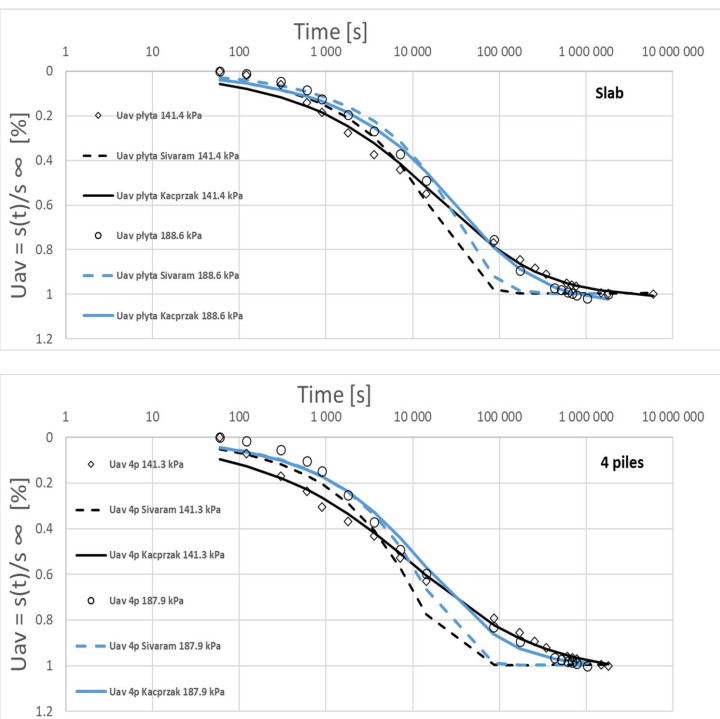

**Figure 10.** *Cont.*

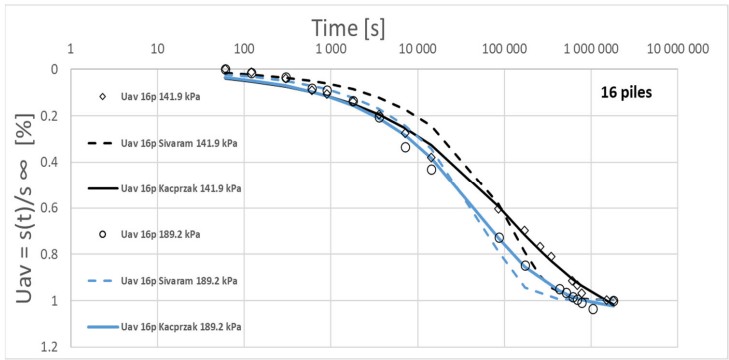

**Figure 10.** Dependence of Uav as a function of time. Comparison of the empirical formula of Sivaram and Swamee (Equation (3)) [21] with test results for 142 kPa and 189 kPa loading. Power coefficients A, B, and C matched to the authors' own test results (5).

**Table 4.** Values of the power coefficients A, B, and C of Equation (5).

| | Slab | | | Slab + 4 Piles | | | Slab + 16 Piles | |
|---|---|---|---|---|---|---|---|---|
| | 142 kPa | | | 142 kPa | | | 142 kPa | |
| A | 0.434 | | A | 0.396 | | A | 0.387 | |
| B | 0.793 | | B | 0.649 | | B | 0.970 | |
| C | 0.543 | | C | 0.601 | | C | 0.366 | |
| | 189 kPa | | | 189 kPa | | | 189 kPa | |
| A | 0.456 | | A | 0.494 | | A | 0.441 | |
| B | 1.117 | | B | 1.014 | | B | 1.320 | |
| C | 0.399 | | C | 0.487 | | C | 0.325 | |

## 5. Preconsolidation Stress σ'c

Performing the interpretation of compressibility curves (stress–settlement relationship), the preconsolidation stress σ'c was determined using the LCPC (Laboratoire central des ponts et chaussées) method for the four types of foundation systems tested (Figure 11).

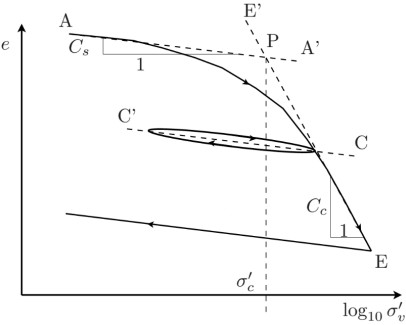

**Figure 11.** Interpretation of preconsolidation stress σ'c on compressibility curve according to the LCPC method.

In accordance with observations during oedometric tests on samples of reconstituted sandy-clay mixtures [24], a convergence of preconsolidation pressure values with no-drain shear strength values was shown. Similar conclusions can be drawn from the current model tests, where the preconsolidation pressure for the soil in the slab test was estimated at 59 kPa, close to the value of the no-drain shear strength measured after the model tests ($s_u$ = 57 kPa—rotary shear, $s_u$ = 64 kPa—piston penetrometer. The preconsolidation stress interpreted from the stress–settlement graphs increases as the number of piles increases (for 4, 9, and 16 piles, σ'c = 87, 94, and 123 kPa, respectively), which can be analogously transposed to an increase in the equivalent shear strength without drainage of the pile-reinforced foundation.

For foundations founded directly at ground level and loaded with permanent loads, the design bearing capacity $R_{FL,d}$ of the subsoil can be calculated using the standard formula in Annex D3 of [2] PN-EN 1997-1:2008 Eurocode 7:

$$\frac{R_{FL,d}}{A\prime} = \frac{(\pi + 2)s_u b_{c,E7} s_{c,E7} i_{c,E7}}{1.4} \tag{7}$$

where

$s_u$—wall strength without drain;
$b_{c,E7}$—slope factor of the foundation base;
$s_{c,E7}$—shape factor of the foundation;
$i_{c,E7}$—load slope factor.

Taking into account the estimated values of equivalent shear strength without drainage, according to the above formula, the bearing capacities of the soil under the slab and slab–pile foundation with 4, 9, and 16 piles are, respectively, 59–5.14–1.2/1.4 = 260 kPa, 87–5.14–1.2/1.4 = 383 kPa, 94–5.14–1.2/1.4 = 414 kPa, and 123–5.14–1.2/1.4 = 542 kPa.

## 6. Numerical Analysis of Large-Dimensional Test Loads of Slab–Pile Foundations

Due to the cost and technical difficulties associated with large loads, large-dimensional load tests are performed very rarely or are limited to the necessary design minimum. Due to the significant cost of preparing research prototypes and the problem of disposing of fabricated samples, the authors decided to prepare a numerical analysis, along with a model calibration, which allows a very precise presentation of the results. Examples include the load testing of a slab foundation and slab–pile system with nine columns [18,24,25]. Complementing the referenced large-scale field studies with test loads for the system with 4 and 16 piles, in addition to the model studies presented in the previous section, the numerical analysis of similar large-scale foundation systems in Plaxis 3D (version v22) software can be used. The scientific objective of this section is to prove the theorem, that settlements of real-scale slab–pile foundations as a function of applied load can be predicted from appropriately scaled model studies [26]. The results of the FEM analysis also allowed us to expand our knowledge of the interaction of slab–pile foundation elements.

### 6.1. Geometric Systems

The testing program included a slab model and three slab–pile models. A 5.0-m-wide square slab was founded directly and on a foundation reinforced with 4, 9, and 16 concrete columns. The relationship between the length of the columns and the width of the slab $L/Br = 2.0$, the identical axial spacing of the columns of 0.8 Br for 4 columns, 0.4 Br for 9 columns, 0.27 Br for 16 columns, as well as the positioning of the piles for the compared systems, with the edge and corner columns at a distance of 0.1 Br from the edge of the slab, were maintained. The authors chose this size of the slab due to the distribution of piles on the slab and the appropriate maintenance of the L/Br ratio. This is determined by the principles presented in Eurocodes and technical studies used in engineering practice. In this system, proportions are important because increasing them will not change the results—the scale effect will be restored. These are the minimum values for the arithmetically assumed pile diameters, where FEM will give an answer as to the behavior of such a system (Figure 12). Convergence with field results confirms the correctly adopted assumptions. The numerical models for all foundation systems assumed that they were made directly at ground level, with no pre-excavation [27].

It is important to mention that the authors did not delve into the presentation of the differential equations needed to achieve results using FEM, including the description of the stiffness matrix aggregation and accompanying discretization activities. Minimal information is provided to focus on comparing the convergence of the numerical analysis with field results. Considering the issue of mathematical description of hardening soil in the operation of stiffness matrix aggregation would be complex and is not the primary purpose of the article.

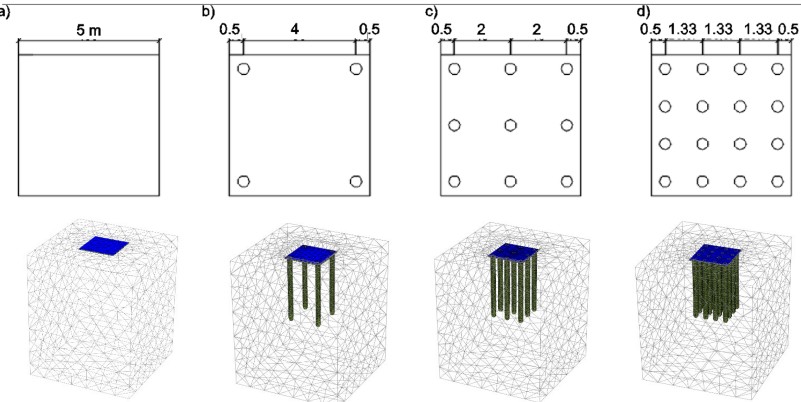

**Figure 12.** Diagram of pile placement under the foundation slab. Slab (**a**) and slab–pile models with 4 (**b**), 9 (**c**), and 16 (**d**) piles.

Finally, it was assumed that the slab model (Table 5) in Plaxis 3D software was founded in soil conditions corresponding to those of the model test sites, i.e., in gyttja described by the constitutive soil model Hs with the parameters given in Tables 5–7.

**Table 5.** Foundation slab parameters.

| $\gamma$ (kN/m³) | Material | $E$ (kN/m²) | $\nu$ | Thickness (m) |
|---|---|---|---|---|
| 0 | Linear, isotropic | $30 \times 10^6$ | 0.2 | 0.3 |

**Table 6.** Parameters of soil medium (gyttia), constitutive soil model Hs.

| $\gamma$ (kN/m³) | Conditions | $e_{initial}$ | $E_{50}^{ref}$ (kN/m²) | $E_{edo}^{ref}$ (kN/m²) | $E_{ur}^{ref}$ (kN/m²) | $p_{ref}$ |
|---|---|---|---|---|---|---|
| 15 | drained | 0.5 | $67.2 \times 10^3$ | $67.2 \times 10^3$ | $134.4 \times 10^3$ | 100 |
| $\nu_{ur}$ | $c$ (kN/m²) | $\phi$ (º) | $\psi$ (º) | $K_0^{NC}$ | Interface | $R_f$ |
| 0.3 | 4 | 18.5 | 0 | 0.68 | 0.67 | 0.9 |

**Table 7.** Concrete column parameters.

| $\gamma$ (kN/m³) | Material | Type | $E$ (kN/m²) | $\nu$ | Interface | Diameter (m) |
|---|---|---|---|---|---|---|
| 24 | Linear isotropic | Poreless | $30 \times 10^6$ | 0.2 | 1.0 | 0.4 |

The concrete columns were modeled as volume piles with the parameters listed in Table 7.

### 6.2. Comparison of Model Test Results with FEM Analysis Results

The results of laboratory-scale model tests of slab–pile systems were compared with the results of numerical modeling of high-dimensional test loads of a slab foundation and three configurations of slab–pile foundations was presented in Table 8 and Figure 13.

**Table 8.** Results of the approximation of load–settlement relationships of CPRF numerical models.

| Type CPRF | Approximate Load Range (kPa) | Approximating Function | Coefficient of Determination $R^2$ |
|---|---|---|---|
| slab | 119–154 | y = 43.795ln(x) − 191.43 | 0.985 |
| slab + 4 piles | 180–250 | y = 44.83ln(x) − 217.92 | 0.962 |
| slab + 9 piles | 254–351 | y = 44.975ln(x) − 234.26 | 0.992 |
| slab + 16 piles | 343–481 | y = 44.435ln(x) − 245.27 | 0.990 |

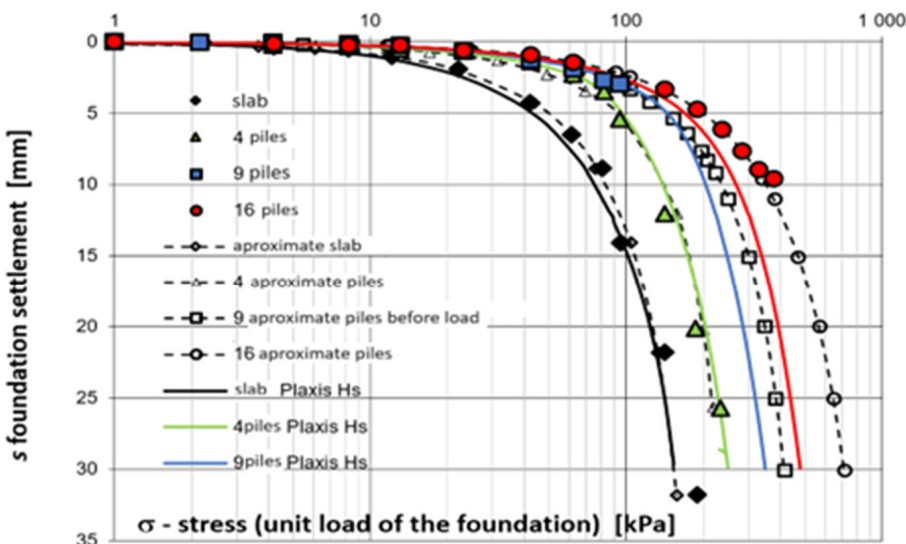

**Figure 13.** Comparison of model test results with FEM analysis results using the Hs model. Stress on a logarithmic scale.

### 6.3. Analysis of Column Behavior Based on FEM Analysis

The columns in the FEA analysis were modeled as volume pile elements with an interface at the nearside [28]. For each of the three steps of forced displacement of the plate, i.e., s = 0.01D, 0.03D, and 0.08D, the distribution of axial force, as well as friction at the lateral along the column, was read (Figures 14–17). The change in axial force as a function of depth was generated automatically using the structural forces in the volume function. A vertical cross-section through the interface of the column's volume element was taken to illustrate the course of friction changes at the sidewall [29].

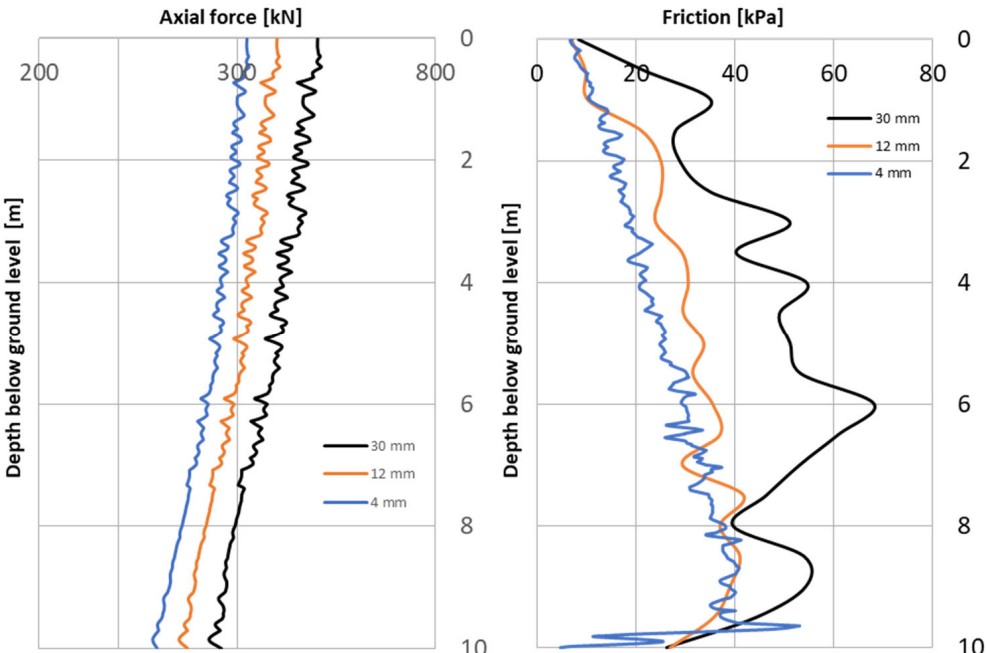

**Figure 14.** Distribution of axial force and friction on the side along the column working alone. Settlement of the column head s = 0.01D, 0.03D, and 0.08D.

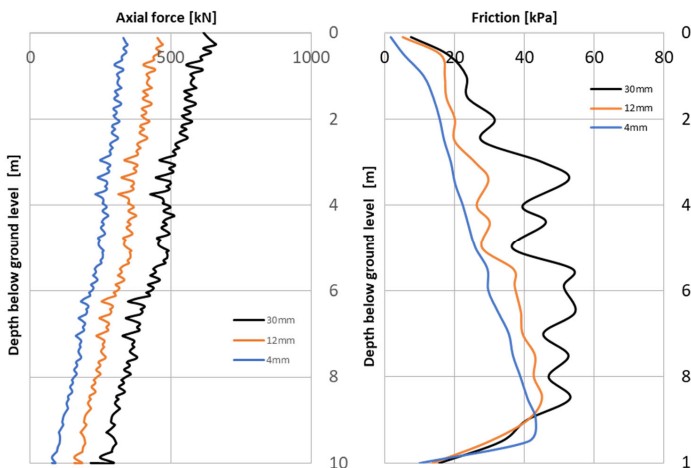

**Figure 15.** Distribution of axial force and friction on the side along the corner column. Plate and 4 columns. Axial spacing of columns r/D = 10. Settlement of column head s = 0.01 D, 0.03 D, and 0.08 D.

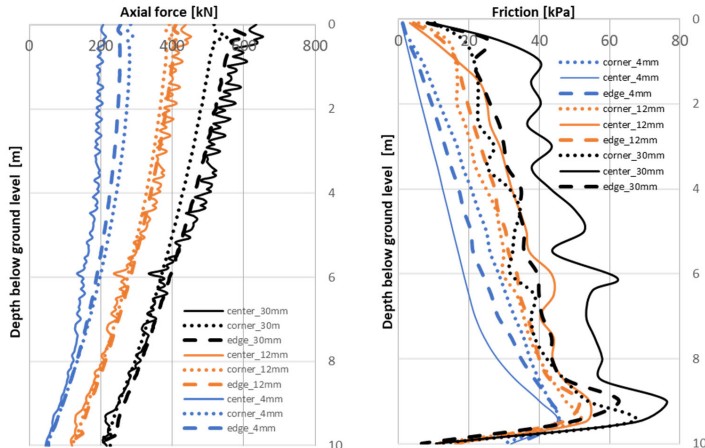

**Figure 16.** Distribution of axial force and friction on the side along the corner, edge, and center columns. Plate and 9 columns. Axial spacing of columns r/D = 5. Column head settlement s = 0.01D, 0.03D, and 0.08D.

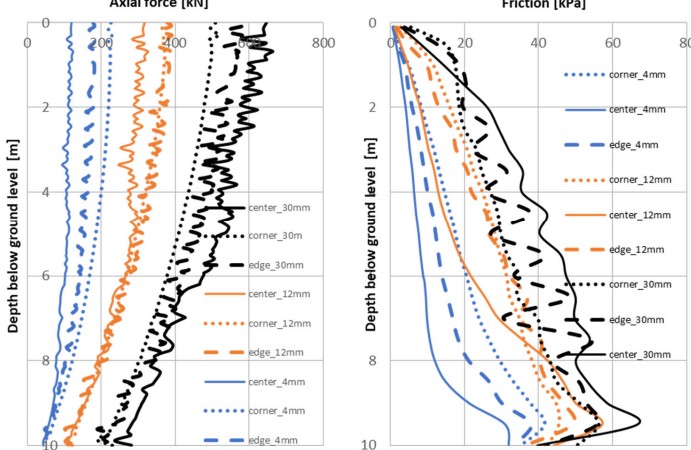

**Figure 17.** Distribution of axial force and friction on the side along the corner, edge, and center columns. Plate and 16 columns. Axial spacing of columns r/D = 3.3. Column head settlement s = 0.01 D, 0.03 D, and 0.08D.

Based on the distributions of axial force and friction on the lateral along the column working independently, as well as being part of a slab and pile system in relation to foundation settlement, presented above (Figures 14–17), it can be concluded that:

- Friction (kPa) on the column (Figures 14–17, figure on the right) increases with depth (the conclusion is valid for a single column working alone as well as for any column, regardless of the number of columns in the pile–raft foundation);
- The greater the settlement (increase in column head settlement from 4 mm to 30 mm), the greater the value of friction mobilized along the column (Figures 14–17, figure on the right, a noticeable increase in friction—smallest values for blue, higher for orange, highest for black lines);
- By comparing the mobilization of friction on the shaft of the column (Figures 14–17, figure on the right) for example for the central columns marked with a continuous blue line, we observe under the slab in pile–raft foundations a reduction in the mobilization of friction on the pile shaft; for example, for a depth of 2 m, the friction (kPa) decreases from 17.1 kPa—single column to 15.8 kPa—column in 4 pile–raft foundation, to 6.4 kPa—column in 9 pile–raft foundation, and finally to 5.88 kPa—column in 16 pile–raft foundation); we observe the so-called formation of a "dead zone" to a depth that depends on the spacing and mutual location of the piles, as well as the amount of settlement of the foundation;
- Reduction of friction on the shaft (based on a comparison of lines of the same color from Figures 14–17) is visible, especially in the range of settlements of the column head of the order of 0.01D (blue line) and 0.03D (orange line);
- For large settlements of the order of 0.08D (black line in Figures 14–17), the vertical pressure of the slab causes a reduction in the dead zone;
- For the range of very small column head settlements (0.01D, blue line), for relative column spacing r/D = 3.3 (Figure 17, figure on the left) to r/D = 5 (Figure 16, figure on the left), the center columns (marked with a continuous line) mobilize less resistance than the edge columns (indicated by a dashed line), or the corner columns (indicated by a dotted line) working most effectively;
- For a range of small column head settlements (0.03D, orange line), for relative column spacing r/D = 3.3 (Figure 17, figure on the left), center columns (marked with a continuous line) mobilize less resistance than edge columns (indicated by a dashed line) or corner columns (indicated by a dotted line);
- For the range of small column head settlements (0.03D, orange line), for relative column spacings r/D = 5 (Figure 16, figure on the left), regardless of their location in the group, the columns mobilize similar resistance;
- For the range of large settlements of the column head (0.08D, black line), for the relative spacing of the columns r/D = 3.3 (Figure 17, figure on the left) to r/D=5 (Figure 16, figure on the left), the center columns (continuous line), due to the significant pressure of the slab, mobilize the greatest resistance; the corner columns (dotted line) work least effectively.

Ultimately, two general conclusions can be recorded: a consequence of the lower value of mobilized resistance is the lower stiffness of CPRF columns, especially in the range of small settlements, and the resistance mobilized by a CPRF column is greater than the resistance mobilized by a column working alone, which is due to the cooperation with the slab.

## 7. Conclusions

The consolidation process is an extremely interesting and important phenomenon that should be considered when designing deep foundations or foundations under high-rise structures. The influence of the consolidation process on the foundation is greater the greater the loads accompanying the project. The whole difficulty is to develop the consolidation factor so that the resulting data set can be used for numerical analysis. This is extremely important from the point of view of the operation of the investment I serviceability limit state. This is because consolidation has a direct impact on the settlement of the object,

and thus the number of defects and inconsistencies during use. The authors presented an analytical description of the consolidation process for slab and pile foundations. This is an interesting take on soil as a construction material that changes its properties over time and should be considered as a system with time-varying parameters. This kind of perspective significantly changes the perception of soil as a material, especially in terms of numerical analyses, for which it is necessary to demand specific parameters. The authors presented a method for calculating consolidation coefficients, the current state of knowledge, and how to apply the obtained results to numerical models. Most importantly, the results obtained in numerical analyses are the same as analytical calculations.

**Author Contributions:** Conceptualization, G.K. and M.F.; methodology, G.K.; software, M.F.; validation, G.K. and M.F.; formal analysis, M.F.; investigation, G.K.; resources, G.K.; data curation, G.K.; writing—original draft preparation, M.F.; writing—review and editing, M.F.; visualization, G.K.; supervision, G.K.; project administration, M.F.; funding acquisition, G.K. All authors have read and agreed to the published version of the manuscript.

**Funding:** This research was funded by Construction Materials Journal.

**Data Availability Statement:** All data used in the article are given in the References section or the authors' own sources have been used, as in the case of photos.

**Conflicts of Interest:** The authors declare no conflict of interest.

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
