# Peer review of "Soil Consolidation Analysis in the Context of Intermediate Foundation as a New Material Perspective in the Calibration of Numerical–Material Models"

_constrmater, doi:10.3390/constrmater3040027_

Round 1

Reviewer 1 Report

Comments and Suggestions for Authors

The consolidation process is an important phenomenon. The authors presented an analytical description of the consolidation process for slab and pile foundations. The study has some significance, however, there are certain issues in the current text that need to be revised.

 1. Currently, the abstract is too wordy and the authors use a lot of content to describe the significance of the results before presenting the results, so it is recommended to consolidate and condense it.

2. The order in which the data is presented in Figure 4 and Tables 1, 2, and 3 seems to be confusing at the moment; according to the author's description, it seems to be 1, 3, 4... ...but the contents of the charts are not in that order.

3. The current numbering of the figures is problematic; the second Figure 6 should be Figure 7, and the order of the figures after that should be changed.

4. The image size should be uniform from Figure 5 to Figure 8. In addition, why most of the fitting curves starting positions and data points are inconsistent?

Author Response

Dear Reviewer,

On behalf of the authors, we would like to thank you very much for your very substantive and professional comments on our article. We are very grateful for your time and for spotting the irregularities in the article. We have read all comments very carefully and have responded to all comments. Thanks to the presented review, our article will gain in quality and clarity.

AD. 1 Thank you for your attention. The abstract has been slightly simplified and consolidated to present the main point of the article.

AD. 2. The order of the columns results from the order in which the results are read from the field research field. We agree that there is no need to follow this order and the readings can be arranged sequentially.

AD. 3. Of course, this is a mistake and was immediately removed. The remaining figures had appropriate numbering, both in the text and in the figure captions, so the current text is now in order. The offset of the drawings was probably created when the publishing house was generating the final file. Thank you.

AD. 4. The authors verified and corrected the graphs illustrating the Meyer curve approximation.

Once again, thank you very much for your comments. According to the authors, we have implemented all the corrections mentioned by the reviewer. We would greatly appreciate you taking the time to review the article.

Best Regards,

Grzegorz Kacprzak

Mateusz Frydrych

Reviewer 2 Report

Comments and Suggestions for Authors

The text is sometimes difficult to understand due to the poor quality of the English language. There are some minor observations and a more substantial one on the lack of information regarding the FEM model. A more in depth analysis of the results would also be desirable.

Lines 28-29

“Before the authors introduce the incredibly interesting issue of soil consolidation in a very special perspective, i.e., with slab and pile foundations”

This seems a bit excessive as an incipit.

Lines 106-107

“in which the global load of the foundation is transferred toS the fSoundation's components”

Please check for unwanted “S”.

Lines 175-177

“The detailed description of the model tests is quite extensive and is the scope of a separate scientific article, which has been accepted for publication and will be published soon.”

Cite the article even if it is not yet published.

Lines 184-185

“the proportion of settlement due to secondary consolidation se”

Do the Authors mean se (“e” in the subscript), rather than se?

Lines 217-218

“As previously mentioned, model studies described in a separate publication by the authors were presented”

This previous publication should be mentioned in References and cited here.

Figure 12

The model appears to have been constructed by giving the ground around the foundation plate a depth of the same order of magnitude as the side of the plate (5m?). This band appears to be too small in size to be able to assume that the soil at the boundaries of the model is in undisturbed conditions, How did the Authors choose this dimension? Have they done sensitivity analyzes when the size of the land band varies? If not, how do they justify the choice made?

Figure 13 (y-axis label)

Please check for unwanted “S”.

Lines 480-507

Each of these conclusions must be justified with reference to the figures presented above. For example:

“directly under the slab in slab-pile foundations, we observe a reduction in the mobilization of friction on the side of the pile”

In which Figure and where in the figure do we observe this reduction?

Comments on the Quality of English Language

Extensive editing of English language required.

Author Response

Dear Reviewer,

On behalf of the authors, we would like to thank you very much for your very substantive and professional comments on our article. We are very grateful for your time and for spotting the irregularities in the article. We have read all comments very carefully and have responded to all comments. Thanks to the presented review, our article will gain in quality and clarity.

The authors did not delve into the presentation of the differential equations needed to achieve results using FEM, including the description of the stiffness matrix aggregation and accompanying discretization activities. Minimal information is provided to focus on comparing the convergence of the numerical analysis with field results. Considering the issue of mathematical description of hardening soil in the operation of stiffness matrix aggregation would be extensive and is not the direct purpose of the article.

Lines 28-29: revised to a simpler sentence.

Lines 106-107: of course it was a mistake, it was immediately corrected.

Lines 175-177: the authors withdrew this statement - currently there is no reference to the unpublished article.

Lines 184-185: Thank you for pointing out this error - it was obviously an editorial mistake, which was immediately corrected.

Lines 217-218: authors add this publication to the bibliography – position [19].

Figure 12: The authors chose this size of the slab due to the distribution of piles on the slab and the appropriate maintenance of the L/Br ratio. This is determined by the principles presented in Eurocodes and technical studies used in engineering practice. In this system, proportions are important because increasing them will not change the results - the scale effect will be restored. These are the minimum values for the arithmetically assumed pile diameters, where FEM will give an answer as to the behavior of such a system. Convergence with field results confirms the correctly adopted assumptions.

Figure 13: it is corrected – s as settlement.

Lines 480-507: The results are presented in Figures 14-17. The authors added a note indicating a reference to the charts.

Once again, thank you very much for your comments. According to the authors, we have implemented all the corrections mentioned by the reviewer. We would greatly appreciate you taking the time to review the article.

Best Regards,

Grzegorz Kacprzak

Mateusz Frydrych

Reviewer 3 Report

Comments and Suggestions for Authors

The submitted manuscript brings original applied research results, thus its publications can be recommended after its careful revision. The details are contained in the attached file.

Comments on the Quality of English Language

The English style is acceptable, but some corrections are necessary: for more details see the last paragraph of the attached file.

Author Response

Dear Reviewer,

On behalf of the authors, we would like to thank you very much for your very substantive and professional comments on our article. We are very grateful for your time and for spotting the irregularities in the article. We have read all comments very carefully and have responded to all comments. Thanks to the presented review, our article will gain in quality and clarity. Thank you.

Below is a reference to the comments presented general remarks:

- The offset of the drawings was probably created when the publishing house was generating the final file.

- Thank you for your attention regarding numbering. This was an obvious mistake, which we immediately corrected. Currently the numbering is correct.

- The text has been edited for correct English. Corrections have been made.

The comments in the .pdf document have been corrected. The reference to the comments has been prepared, addressing each comment in turn. For clarity, corrected comments from dashes have been listed:

- Introduction edited, missing brackets and references corrected.

- reference no. [15] was checked and we confirm that it is corrected

- the symbol for pre-consolidation stresses has been corrected

- the values in table 6 have been corrected.

- removed repeated or unclear words in the text

Regarding numerical analysis:

- the authors decided that introducing the differential form of equations into the article is not necessary in this approach, because these are developed issues that have been implemented into the software and in this specific article, and the authors focused on the results of the method, i.e. the comparison of numerical analysis with the experiment performed in reality

- the readability of charts has been improved

- the bibliography has been corrected

- the drawings have been corrected

- added spaces in appropriate places

- the abstract has been corrected

- due to capital letters these are obvious mistakes that were immediately corrected

Once again, thank you very much for your comments. According to the authors, we have implemented all the corrections mentioned by the reviewer. We would greatly appreciate you taking the time to review the article.

Best Regards,

Grzegorz Kacprzak

Mateusz Frydrych

Round 2

Reviewer 2 Report

Comments and Suggestions for Authors

The Authors addressed the comments and improved the manuscript, but some changes are still needed.

Lines 213-215

“As previously mentioned, model studies described  in a separate publication by the authors were presented to analyze settlement over time of slab-on-grade foundations [19].”

Since you deleted the sentence in which you already talked about the model (“The detailed description of the model tests is quite extensive and is the scope of a separate scientific article, which has been accepted for publication and will be published soon.”), you should delete “As previously mentioned”.

You should also add this part of your explanation in the text:

“The authors did not delve into the presentation of the differential equations needed to achieve results using FEM, including the description of the stiffness matrix aggregation and accompanying discretization activities. Minimal information is provided to focus on comparing the convergence of the numerical analysis with field results. Considering the issue of mathematical description of hardening soil in the operation of stiffness matrix aggregation would be extensive and is not the direct purpose of the article.”

Figure 12

“The authors chose this size of the slab due to the distribution of piles on the slab and the appropriate maintenance of the L/Br ratio. This is determined by the principles presented in Eurocodes and technical studies used in engineering practice. In this system, proportions are important because increasing them will not change the results - the scale effect will be restored. These are the minimum values for the arithmetically assumed pile diameters, where FEM will give an answer as to the behavior of such a system. Convergence with field results confirms the correctly adopted assumptions.”

Add this explanation to the text.

Lines 475-502

The reviewer is still of the opinion that each bullet point should refer to its figure.

Comments on the Quality of English Language

Extensive editing of English language required.

Author Response

Dear Reviewer,

On behalf of the authors, we would like to thank you for your valuable comments. We have addressed all of them, changing the text of the paper accordingly. Thank you very much for your time.

Lines 213-215: On behalf of the authors, we would like to thank you very much for this attention. We have edited the text accordingly and added a short explanation about FEM.

Figure 12: Thank you. Of course, the authors added an explanation to the text, editing it so that it was readable for the recipient.

Lines 475-502: The authors rearranged the summary by describing what is presented in the charts and how to interpret it. Attached is the corrected text of the article.

Thank you for your great help and very valuable comments.

Kind Regards,

Grzegorz Kacprzak

Mateusz Frydrych
